# Physiology and Molecular Breeding in Sustaining Wheat Grain Setting and Quality under Spring Cold Stress

**DOI:** 10.3390/ijms232214099

**Published:** 2022-11-15

**Authors:** Hui Su, Cheng Tan, Yonghua Liu, Xiang Chen, Xinrui Li, Ashley Jones, Yulei Zhu, Youhong Song

**Affiliations:** 1School of Agronomy, Anhui Agricultural University, Hefei 230036, China; 2School of Horticulture, Hainan University, Haikou 570228, China; 3Research School of Biology, The Australian National University, Canberra, ACT 2601, Australia; 4Centre for Crop Science, Queensland Alliance for Agriculture and Food Innovation, The University of Queensland, Brisbane, QLD 4072, Australia

**Keywords:** *Triticum aestivum* L., spring frost, spikelet development, grain set and quality, QTLs

## Abstract

Spring cold stress (SCS) compromises the reproductive growth of wheat, being a major constraint in achieving high grain yield and quality in winter wheat. To sustain wheat productivity in SCS conditions, breeding cultivars conferring cold tolerance is key. In this review, we examine how grain setting and quality traits are affected by SCS, which may occur at the pre-anthesis stage. We have investigated the physiological and molecular mechanisms involved in floret and spikelet SCS tolerance. It includes the protective enzymes scavenging reactive oxygen species (ROS), hormonal adjustment, and carbohydrate metabolism. Lastly, we explored quantitative trait loci (QTLs) that regulate SCS for identifying candidate genes for breeding. The existing cultivars for SCS tolerance were primarily bred on agronomic and morphophysiological traits and lacked in molecular investigations. Therefore, breeding novel wheat cultivars based on QTLs and associated genes underlying the fundamental resistance mechanism is urgently needed to sustain grain setting and quality under SCS.

## 1. Introduction

Wheat provides approximately 20% of the food energy and protein produced for human consumption [1], and grain quality is an important indicator due to market value and consumer acceptance [2,3]. Wheat grain quality is a complex combination of various traits, mainly controlled by genotypic and environmental factors [4]. Climate change is causing a temperature shift and ecological landscapes that negatively impact wheat yield and quality [5]. During the last several decades, it has been reported that spring cold stress (SCS) has caused severe losses in wheat production and grain quality. For example, in Australia, the SCS events that frequently occurred at wheat reproductive stage typically resulted in yield losses of 10%, and it’s more than 85% in various farmlands [6,7]. Nearly 85% of China’s total area planted with winter wheat experiences widespread SCS [8,9]. Reports from North America and Europe indicated that late frost spells are one of the most economically damaging agricultural climate hazards, causing substantial economic losses in 2017 [10,11]. Consequently, the abiotic stress of SCS threatens the safety of crop production systems worldwide. Wheat growth and development have been subjected to more frequent cold stress as climate change continues [12].

The SCS events often occur during the reproductive development in winter wheat [13]. The reproductive development is composed of floral initiation, pollen grain and embryo development, pollination, fertilization and grain setting, etc. [14]. When wheat suffers from frost during the reproductive growth period, it causes the wheat spike cells to lose water and wither, affecting the young spike’s normal development and increasing the young spike’s mortality [15,16]. Malfunctions and irreversible abortion of male and female reproductive organs and gametophytes are the main reasons for cold-induced male and female infertility [17]. During SCS, the anthers display irregular hypertrophy and vacuolation of the tapetum, an unusual accumulation of starch and protein in the plastids, and poor pollen tube development [18,19]. Zhang et al. (2021) stated that low-temperature stress significantly reduced the expression and activity of the sucrose invertase (CWINV) coding gene in young ears at the booting stage, inhibited the transport of sucrose to pollen sac, and then hindered the normal development of pollens [20]. Occurrence of SCS at late reproductive growth resulted in smaller dark-colored seeds with a wrinkled epidermis, poor seed setting and quality [21].

Wheat responds to cold stress by regulating key physiological, biochemical, and molecular mechanisms [22]. Under cold stress, a wide range of chemicals or protective proteins are produced, including soluble carbohydrates, proline, and cold-resistance proteins [23], which are involved in regulating osmotic potential, preventing ice crystal formation, the stability of cell membranes and reactive oxygen species (ROS) scavenging [24]. At the molecular level, estimates of phenotypic plasticity were used to identify loci associated with stress tolerance. Candidate genes involved in phytohormone-mediated processes for stress tolerance were proved to be involved in cold stress responses [25]. Cold acquisition of freezing tolerance requires the orchestration of disparate physiological and biochemical changes, and these changes are mainly mediated through the differential expression of genes [26,27]. Some of these genes encode effector molecules directly involved in stress mitigation, and others encode proteins for signal transduction or transcription factors that control gene pool expression [26]. Genes involved in plant metabolism were differentially expressed to avoid injury and damage associated with SCS; it includes the encodings of Ca^2+^ binding proteins, protein kinases, and inorganic pyrophosphatase [28].

Understanding the potential regulatory mechanisms behind SCS tolerance is necessary to create wheat breeding varieties with improved grain setting and quality under cold stress. In this review study, we further summarized the consequences of SCS and explored the potential mechanisms to sustain wheat grain setting and quality under SCS. The objectives of this study are to (i) make clear the physiological and molecular mechanism in controlling grain setting and quality under SCS, and (ii) propose breeding strategies in combatting SCS during reproductive stage.

## 2. Effects of SCS on Grain Number and Quality in Wheat

Under varying climatic conditions, the SCS events have become more frequent, intense, and prolonged. The SCS events often occur during the reproductive stage of winter wheat, which is critical for the establishment of the panicle [29]. The SCS compromises the development of young spike and floret; nutrient distribution is altered, and floret stunting (or sterility) occurs, resulting in poor grain set and quality (Figure 1).

### 2.1. Grain Number

Grain number is a significant factor in determining wheat grain yield [30]. The stages from jointing to flowering are critical to prevent florets from degenerating and increase the grain setting rate [31,32,33]. Under SCS conditions, the lower spike number per plant and grain number per spike were primarily responsible for reduced grain production (Table 1; Figure 2) [34]. Compared with spring wheat cultivars, semi-winter wheat has stronger cold resistance. For example, under low temperature of −2–6 °C for 3 days at the jointing stage, the grain number per spike was lowered by 1.3–4.4% in Yangmai16 (spring wheat), while decreased by 0.6–1.0% in Xumai30 (semi-winter wheat) [35]. Meanwhile, cold stress led to different yields of different genotypes at the reproductive stage [36]. Compared with the control, low temperature led to zero harvest of diploid genotypes, and the yield of tetraploid genotypes decreased significantly, while hexaploid genotypes acquired relatively high maintenance rate of grain yield among three species [36]. Additionally, the yield loss caused by SCS also depends on the intensity of the low temperature and its duration [37]. Ji et al. (2017) exposed two wheat cultivars at the booting stage to freezing temperature at 2, −2, −4 and −6 °C for 2–6 d in a convective freezing chamber, causing 13.9–85.2% grain yield reduction in spring wheat, while resulting 3.2–85.9% grain yield loss in semi-winter wheat [35]. With the temperature declined to −5 °C and −7 °C at the vegetative growth stage, the grain yield decreased by 10–100% [38]. In each case, the SCS events during the reproductive development significantly affected the growth and development of younger spikes and florets, causing pollen infertility and poor grain setting [39], thereby resulting in a decrease in the number of grains.

### 2.2. Grain Quality

Grain quality is primarily based on appearance and nutritional quality [43]. It is well known that mostly spring cold stress events are often encountered during the reproductive period in wheat, which seriously affects the absorption and distribution of nutrients [39]. Grain quality relative to its appearance refers to external morphological characteristics, including grain length, width, and aspect ratio [44]. For example, wheat responds to SCS (−4 °C for 12 h) at the jointing stage by increasing the ratio between grain length and width (L/W) for 0.4–14.2% while decreasing the equivalent diameter in 0.9–11.0% and grain area in 1.6–20.2% [45]. Compared to the cold-tolerant genotype, the grain width and L/W of the sensitive wheat genotype were more susceptible to low temperatures [46]. It also reported that the grain width is more sensitive to low temperatures than the grain length [46].

In addition to affecting morphological appearance, the quality of grain nutrition is adversely affected by SCS [47]. For wheat grain nutrition quality, protein content is of key significance [48]. It has been noted that SCS limits the production of nitrogen compounds and nonstructural carbohydrates, which decreases the transit of protein and total soluble sugar from stems into grains, resulting in a decline in wheat quality [36]. Under low temperatures at the booting stage, the mean accumulation of total protein decreased by 4.8–6.9%, albumin by 5.8–9.6%, globulin by 8.4–15.4%, gliadin by 13.2–18.4%, and glutenin by 17.8–29.1% [49]. In addition to this, reductions in the concentrations of amylose, amylopectin and total starch were also observed under different low-temperature levels [46]. According to a recent report, the total starch in wheat grains, as well as the rate of accumulation of straight-chain and branched-chain starch, were closely related to the activities of starch branching enzyme (SBE), soluble starch synthase (SSS), granule-bound starch synthase (GBSS) and adenosine diphosphate glucose pyrophosphorylase (AGPase) [50], while the activity of essential starch synthesis enzymes is particularly sensitive to SCS during grain development [51]. The low temperature during the reproductive stage decreased the activities of crucial starch synthesis enzymes (AGPase, SSS, GBSS, and SBE) in the grain, thereby reducing the accumulation of starch, resulting in a decreasing grain quality [52].

## 3. Physiological Mechanism of Controlling Wheat Resistance to Cold Stress

### 3.1. Protective Enzymes for Oxidation

Cold stress often leads to excess accumulation of reactive oxygen species (ROS) such as superoxide radical (O^2−^) and hydrogen peroxide (H_2_O_2_), which causes oxidative damage to DNA, proteins, and lipids, leading to the inhibition of wheat seed development [53,54]. Hence, the balanced ROS production level was achieved at the intracellular level which promotes the normal growth, development, and cellular metabolism (Figure 3) [55].

The activation of subcellular antioxidant mechanisms can provide some resistance to SCS in wheat while also decreasing oxidative burst in the photosynthesis machinery [56]. Activities of antioxidant enzymes, such as peroxidase (POD), superoxide dismutase (SOD), and catalase (CAT), play an essential role in protecting plants from oxidative damage by ROS scavenging [57,58]. Several studies have reported that alterations in the activity of numerous antioxidant defense system enzymes help plants to handle oxidative stress in wheat [59,60]. For example, cold stress (4 °C and −4 °C) increased the activity of SOD by 6.8–68.3%, POD by 16.6–69.4%, CAT by 6.0–53.8% in a wheat spikelet, compared to optimum temperature (16 °C) [61]. Furthermore, antioxidant chemicals, including proline, glutathione (GSH) and ascorbic acid (AsA), also play critical roles in protecting plants from ROS damage caused by cold stress [62]. Under SCS, the accumulation of proline eliminates oxygen free radicals, which balances the osmotic pattern in the cell, and maintains the normal state of the membrane [63]. For example, the application of exogenous proline improved wheat’s cold tolerance, due to the increased accumulation of free proline and sucrose, by coordinating carbon and nitrogen metabolism [64]. It is noted that the AsA–GSH cycle, including ascorbate peroxidase (APX), monodehydroascorbate reductase (MDHAR), dehydroascorbate reductase (DHAR), and glutathione reductase (GR), are very effective in improving wheat cold tolerance, particularly to ROS stress [65]. For example, AsA could induce the up-regulation of diverse antioxidants (super oxide dismutase (SOD), peroxidase (POD), and catalase (CAT)), thus offsetting the adverse effects of cold stress at early and reproductive stages of wheat [66].

Another key mechanism in plant cold stress responses is the regulation of transcription by endogenous hormones and ROS [67]. Once induced by cold stress, hormones change the ROS levels due to increasing transcription or talking about post-translational modification/activation of proteins and transforming ROS signaling [68]. For instance, it has been demonstrated that the ROS generated by RBOHs mediates an interaction between ABA and BRs, enhancing cold tolerance in *Arabidopsis* [69]. According to a recent study, the application of exogenous BRs increased antioxidant capability, directing the reduction of oxidative damage caused by ROS bursts [70].

### 3.2. Carbohydrate Metabolism

Carbohydrate metabolism plays an essential role in energy availability for plant development and also has a role in temperature acclimation [71]. In plants, several soluble sugars, such as sucrose, glucose, sucrose, fructose, raffinose and trehalose, act as biofilm protectors by interacting with the lipid bilayer. This interaction has a role in reducing membrane damage, as the sugars function as osmoprotectants and provide adaption to the cold environment [72,73].

The soluble sugars sucrose, glucose, trehalose, and fructose start accumulating in response to cold stress, enhancing cold tolerance during the reproductive stage of crops [74]. For instance, the buildup of soluble sugars under SCS can raise the amount of proline, which controls osmotic pressure, scavenges reactive oxygen species, and stabilizes biomolecule structure, reducing low-temperature damage [75,76]. Fructans, which are highly water soluble, act as osmoregulatory substances to prevent the formation of ice crystals in the cytoplasm and improve biofilm stability, enhancing crop cold tolerance [77]. Recent research has confirmed a high correlation between fructan accumulation and cold tolerance due to increasing transcript levels of the Cor (cold-responsive)/Lea (late-embryogenesis-abundant), C-repeat-binding factor (CBF), and fructan biosynthesis-related genes in the wheat family [77]. Trehalose has been found to act as an osmoprotectant, and stabilizes protein integrity in plants [78]. Importantly, exogenous trehalose prevented floret degeneration under low-temperature conditions and increased floret fertility in young spikelets, minimized any loss in grain number per spike [43].

Recently, the Sugars Will Eventually be Exported Transporters (SWEETs) have been reported to regulate abiotic stress tolerance, sugar transport, plant growth and development [79]. The SWEETs also play vital roles in oxidative and osmotic stress tolerance [80]. In wheat, the genome-wide analysis revealed 105 SWEETs, and 59% exhibited significant expression changes under abiotic stresses [81]. Importantly, *AtSWEET16* and *AtSWEET17* are two bidirectional vesicular fructose transporters that maintain glycan homeostasis and promote the accumulation of fructose in vacuoles, which may be beneficial in stress tolerance responses [82,83]. A further understanding of sugar metabolism and transport will be key in reducing any sugar starvation in crop reproductive development and enhancing seed setting rate.

### 3.3. Hormones and Ca^2+^ Signals

Plants adapt to environmental changes in low-temperature settings by a sequence of cellular reactions triggered by signaling molecules (e.g., hormone signals, Ca^2+^ signal), which result in plant defense and adaptability to adverse conditions [84,85]. Plant hormones, such as abscisic acid (ABA) [86], jasmonic acid (JA) [87], and salicylic acid (SA) [88], have been reported to play a significant role in regulating grain quality. Past findings revealed that many plants experience higher endogenous ABA levels in response to cold stress [89,90]. In wheat, the application of exogenous ABA is reported to enhance cold tolerance by increasing the activities of antioxidant enzymes and reducing H_2_O_2_ contents under cold stress [91]. In particular, ABA-dependent gene expression, which includes the ABA receptors, protein phosphatases type-2C (PP2Cs), Snf1-related kinase 2s (SnRK2s), and AREB/ABF regulon, controlled by the raised ABA levels, helped plants adapt to abiotic stress cold stress [92]. According to Zhang et al. (2018), the significant up-regulation of the *SnRK2.11*, serine/threonine-protein kinase and serine/threonine-protein phosphatase PP1-like was considered to be a significant reason for improving cold tolerance in wheat during the reproductive stage [28]. These genes were believed to function in ABA signaling in guard cells.

Additionally, JA also plays a mediating role in synthesis and signaling to mediate low-temperature tolerance [93]. For instance, endogenous JA levels were found to be increased in wheat [94], rice [95], and Arabidopsis [96], enhancing the frost resistance of crops. JA functions as an upstream signal of the ICE-CBF pathway, positively modulating freezing responses [97]. *JAZ1* and *JAZ4* are JA signaling negative regulators interacting with ICE1 and ICE2 to repress their transcriptional activity [98]. Subsequently, they regulate the expression of CBF and other low-temperature responsive genes, thus affecting wheat cold resistance [97].

It is well known that SA plays a vital role in responding to abiotic stresses, apart from regulating crop growth, ripening and development [98,99]. SA activates the active oxygen species before low-temperature exposure; it promotes an increase in antioxidant enzyme activity and higher mRNA content of *TaFeSOD*, *TaMnSOD*, *TaCAT* gene transcripts, and free Proline after SCS [100]. Freezing stress during the reproductive stage shows salicylic acid-primed wheat up-regulated the expression level of the WRKY gene (*WRKY19*), heat shock transcription factor (*HSF3*), mitochondrial alternative oxidase (AOX1a), and heat shock protein (HSP70), which contributes to increasing of antioxidant capacity and protection of photosystem in parallel with lower malonaldehyde content, superoxide radical production as compared with non-primed wheat [101]. Further research has demonstrated that SA treatment reduces ice nucleate and induces anti-freezing protein, which inhibits the formation of ice crystals in plant cells [88].

Ca^2+^ is an essential secondary messenger in plants in response to cold stress [102]. Ca^2+^ sensors such as calmodulins (CaMs), CaM-like proteins (CMLs), Ca^2+^-dependent protein kinases (CPKs/CDPKs), and calcineurin B-like proteins (CBLs) are the primary transmitters of the Ca^2+^ signal that is induced by cold stress [103,104,105]. For example, *OsCPK27*, *OsCPK25*, and *OsCPK17* activated MAPK, ROS, and nitric oxide pathways in response to cold stress [85,106]. Recently, genome-wide identification and expression analysis also show that 18 *TaCaM* and 230 *TaCML* gene members were identified in the wheat genome, and *TaCML17*, *21*, *30*, *50*, *59* and *75* were identified related with responses to cold stress in wheat [107].

### 3.4. Transcription Factors

Wheat genomes contain a large number of transcription factors that play important roles in cold-stress biological processes, including CBF [108], basic leucine zipper (bZIP) [109], MYB [110], and NAC [111].

The ICE-CBF-COR signaling pathway is widely recognized as essential for cold adaptation [112]. The receptor protein detects cold stress and initiates signal transduction, activating and regulating the ICE gene, which up-regulates the transcription and expression of the CBF gene [113]. In wheat, five ICE genes, 37 CBF genes and 11 COR genes were discovered in the wheat genome database [114]. Wheat CBF genes have been demonstrated to improve cold tolerance in other plants, as shown with transgenic barley containing *TaCBF14* and *TaCBF15* genes [115]. A vast variety of transcription factors are also important, such as CBF1, CBF2, and CBF3 [116] and C-type repeats (CTR) [117], which play crucial roles in the biological processes of abiotic stressors in wheat. Previous studies reported that cold-regulated transcriptional activator CBF3 positively regulates cold stress responses in wheat [118]. The RNA-seq data and qRT-PCR revealed that the ICE, CBF, and COR genes have varying expression patterns in different wheat organs, with ICE genes mainly up-regulated in the grain, CBF in the root and stem, and COR in the leaf and grain [114]. All these results show that the ICE–CBF–COR cascade plays a crucial role in the response of wheat to cold stress (Figure 3).

The bZIP genes are involved in important regulatory processes of plant growth and physiological metabolisms, such as promoting anthocyanin accumulation [119] and other signals [120]. Similarly, the bZIP gene also has a variety of biological functions under abiotic stress, and 187 bZIP genes have been predicted in wheat [121]. And the majority of bZIPs linked to frost tolerance in plants are positive regulators [122]. For example, phenotypic analysis and related physiological indicators of cold resistance showed that overexpression of *TaABI5* could enhance cold resistance [109]. In recent years, 15 bZIP genes with variable expression were found in early wheat spikes, and most showed an increase in in expression in response to SCS [123]. Furthermore, the bZIP genes are involved in ABA signaling and play a role in responding to freezing stress in the later stage of wheat [109]. Similarly, MYB and NAC are crucial in controlling plant growth and cold stress responses [110,124].

## 4. Breeding Strategies to Develop SCS-Resistant Wheat

Superior wheat genotypes are needed for SCS resistance, which will be made possible by breeding cold-resistant cultivars that maintain yield stability and high quality [125]. Appropriate measures need to be taken to cope with the consequences of SCS in wheat during the reproductive stage, to improve crop yield and quality. Strategies to strengthen SCS resistance include selecting cold-tolerance cultivars, identifying QTL/genes, and exploiting closely linked markers in wheat.

### 4.1. QTLs Associated with Cold Resistance

Genetic components such as QTLs have great potential to accelerate traditional breeding processes [126]. QTLs related to cold tolerance and the underlying molecular mechanisms have been thoroughly studied in wheat [127,128]. There are loci for cold resistance on 1B, 1D, 2B, 2D, 4D, 5A, 5D, and 7A, with 5A and 5D suspected to carry significant genes of interest [129,130]. Wheat chromosome 5A plays a key role in cold acclimation and frost tolerance [119]. Three key genes responsible for SCS tolerance, *Fr-1* (e.g., *Fr-A1*, *Fr-B1*) and *Fr-2* (e.g., *Fr-D1*), were located on chromosomes 5A, 5B, and 5D [131,132], with two loci being mapped within a distance of approximately 30 cM [118]. The *Fr-1* maps close to the vernalization locus Vernalization-1 (*Vrn-1*), so they showed highly homologous [133]. The *Vrn1* acts as a positive regulator of vernalization and regulates the transition from vegetative to reproductive growth in wheat [134]. The *Fr-A^m^2* locus is made up of a group of eleven CBF genes that are activated during vernalization, which in turn activate the COR genes necessary for enhanced cold tolerance of wheat [135,136].

Genome-wide association studies (GWAS) of traits related to wheat resistance and tolerance are essential to understanding their genetic structure for improving breeding selection efficiency [137]. 23 QTL regions located on 11 chromosomes (1A, 1B, 2A, 2B, 2D, 3A, 3D, 4A, 5A, 5B and 7D) were detected for frost tolerance in 276 winter wheat genotypes by GWAS, eight novel QTLs were discovered on chromosomes 1B, 2D, 3A, 3D, 4A and 7D [129]. Eighty SNP loci distributed in all the 21 chromosomes were associated with the resistance of SCS with 16.6–36.2% phenotypic variation by GWAS, six loci of these were stable loci with more than two traits, and multiple superior alleles were obtained from the associated loci related to SCS traits [138]. Nevertheless, the majority of the QTL intervals for low-temperature tolerance reported by GWAS are still huge, and there are too many candidate genes; the causal genes for cold tolerance are still challenging to find.

Of the different genome editing approaches, CRISPR/Cas9 genome editing module has evolved as a successful tool in modulating genes essential for developing high-stress resistance of crops [139]. Meanwhile, CRISPR/Cas9 allows the manipulation of the wheat genome for improved agronomic performance, resistance to biotic and abiotic stresses, higher yields, and better grain quality [140]. For example, Tian et al. (2013) [141] cloned *TaSnRK2.3*, then further determined its expression patterns under freezing stresses in wheat emerging and characterized its function in Arabidopsis. Overexpression of *TaSnRK2.3* significantly enhanced tolerance to freezing stress, enhancing the expression of cold stress-responsive genes and ameliorating physiological indices [141]. Additionally, it showed that overexpressing *TaFBA-A10* led to the increased activity of FBA, as well as regulating key enzymes in the Calvin cycle and the glycolysis rate to enhance cold tolerance of wheat [142]. Therefore, acceptance and utilization of new plant breeding technologies involving genome editing confer opportunities for sustainable agriculture and ensure global food security.

### 4.2. Cultivars for SCS Resistance Based on Agronomic Traits

Wheat yield is associated with several agronomic traits which have been used to make better cultivars, increasing the yield and quality of wheat [143]. Given the high heritability of the traits and the relevance of wheat yield, agronomic traits can be used as selection criteria in breeding and cultivar development (Table 2) [144]. Cold stress affects agronomic traits at every developmental stage, but the reproductive stages are relatively more sensitive [145]. Specifically, cold stress affects the development of young spikes and flowers, grain characteristics and quality [146,147]. Some researchers have classified inversions into five major categories based on the degree of damage to the spikelet: grade 1 for no apparent frost damage, grade 2 for frost damage less than 1/3, grade 3 for frost damage between 1/3 and 1/2, grade 4 for frost damage greater than 1/2, and grade 5 for all young spikes that died from freezing [148]. Similarly, frost damage also impaired stem development, resulting in lower plant height and a decrease number of spikes [149]. For example, using the dead stem rate to classify 120 wheat cultivars into five classes of very strong, moderate, weak and very weak, and determining the criteria for categorizing wheat spring frost resistance evaluation classes [150].

Moreover, biomass accumulation is also a significant source of grain yield and a growth process sensitive to cold stress [150]. The SCS has adverse effects on several wheat metrics, including the mean leaf area index (MLAI), mean net assimilation rate (MNAR), harvest index (HI), biomass per plant (BPPM), and grain yield per plant (GYPP) [35]. These metrics can be utilized in wheat breeding programs to assist in developing cold-tolerant varieties.

### 4.3. Cultivars for SCS Resistance Based on Molecular Traits

It is critical for breeding to understand the physiological features linked to genetic improvements in yield and quality [153]. When SCS harms wheat, a variety of complicated physiological and biochemical changes take place inside the plant that has an impact on yield and quality. Reactive oxygen [154], MDA content, antioxidant enzyme activity [56], carbohydrates [155], osmoregulatory substances [87], hormone content [91], starch content [156], and photosynthesis [157] are often used as physiological and biochemical indicators for wheat inversion identification (Table 2). According to Zhang et al. (2019), the quantity of wheat-bearing grain can be considered to determine POD activity, SOD activity, and MDA level as indices of wheat cold resistance [158]. To determine the extent of freezing damage, Wang et al. (2022) used principal component-affiliate function-stepwise regression analysis to screen seven important physiological indicators: chlorophyll a, leaf water content, proline, Fv/Fm, soluble protein, MDA, and SOD. The equation coefficient of determination between the predicted value of the integrated index of freezing damage and yield established from this reached 0.898 [159]. Following an abrupt temperature drop, it was discovered that in cold-tolerant wheat cultivars, the expression of genes encoding antioxidant enzymes increased, antioxidant enzyme activity was improved, and ROS content was decreased, whereas ROS content was higher, and some leaves died in cold-sensitive wheat cultivars [160].

To enhance wheat tolerance to SCS and improve sustainability, many researchers focus on understanding the key molecular targets, regulatory pathways and signaling designed for genotype–environment interactions [161,162]. As an important research tool for functional genes, transcriptome sequencing has been employed in regulatory network investigations of plants under abiotic stress [163]. In wheat, 450 genes were found to have altered transcript abundance following 14 low-temperature treatments, including 130 candidates for transcription factors, protein kinases, ubiquitin ligases, GTP, RNA, and Ca^2+^ binding proteins genes [164]. Transcriptome sequencing of cold stress during reproductive stages in wheat identified 562 up-regulated, and 314 down-regulated differentially expressed genes, and these genes were mainly involved in photosynthesis, lipid and carbohydrate synthesis, amino acid and protein accumulation [165]. According to transcriptomics and metabolomics analysis, the ABA/JA phytohormone signaling and proline biosynthesis pathways play an important role in regulating cold tolerance in wheat [94]. Transcription is only part of the response; many researchers also employ proteomics for in-depth analysis of protein changes, offering global analysis of protein accumulation [166]. Proteomic analysis has been carried out in wheat under SCS [167], with various proteins being identified as having a role in cold tolerance, providing protection against cold damage [168]. For instance, the proteomic analysis of wheat under low temperatures revealed an upregulation of the expression of proteins involved in signal transduction, carbohydrate metabolism, stress and defense responses, and phenylpropane biosynthesis [169].

## 5. Conclusions and Future Perspectives

SCS incidents more often occur under changing climatic conditions, causing a serious threat to wheat reproductive tissues and grain production. The SCS is detrimental to the development of the floret and spikelet in wheat; thus, compromising the grain number and quality. A premium cultivar tolerating SCS is a prerequisite for sustaining wheat farming. The review shows that the protection of young, tender spikelet issues in wheat from cold stress impacts was mainly dependent on the collective contribution of antioxidant enzyme activity, carbohydrate accumulation, hormone signaling and transcriptional regulation. The effort of breeding cultivars with simple agronomic and morpho-physiological traits has been made in coping with cold stress, which should be improved by identifying novel SCS-tolerant QTLs or genes with regards to floret and spikelet development in new breeding strategies which embrace fundamental mechanisms. Further studies on multi-omics, from genomics to phenomics, to identify the genes regulating cold tolerance will be necessary for future breeding programs.

## Figures and Tables

**Figure 1 ijms-23-14099-f001:**
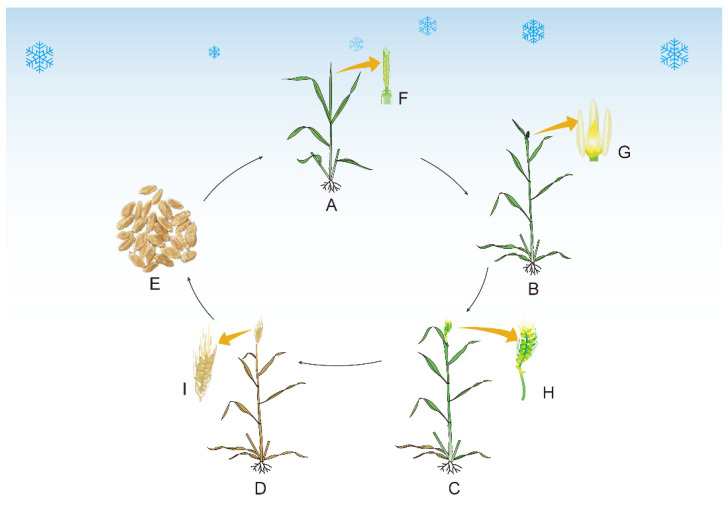
A schematic diagram visually demonstrating the impacts of SCS initiated at tetrad stage (**A**) on subsequent wheat growth and development at booting stage (**B**), anthesis stage (**C**) and maturation stage (**D**). (**F**) Indicates young spikelet development at tetrad stage. (**G**) Indicates tapetum degeneration and pollen sterility in the developing anthers at booting stage. (**H**) Indicates reduced pollen viability and thus spikelet fertility. (**I**) Indicates reduced grain-filling rate and period and enhanced grain abortion, and thus less grain number and quality (**E**).

**Figure 2 ijms-23-14099-f002:**
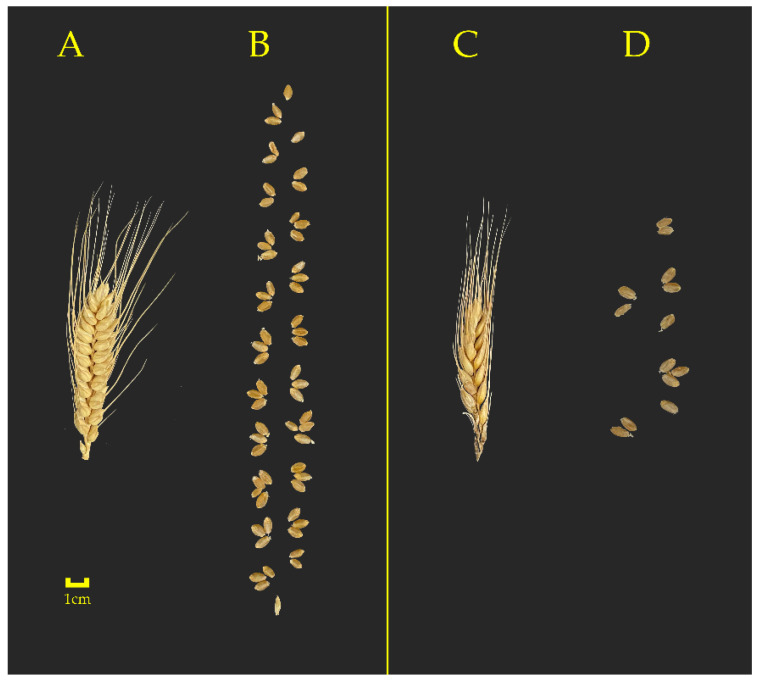
The photos of spikes (**A**,**C**) and grains (**B**,**D**) in wheat under the control (**A**,**B**) and spring cold stress (**C**,**D**) −2 °C for 6 h. The photo visually shows the effect of cold stress on the size and color of spike and grain number.

**Figure 3 ijms-23-14099-f003:**
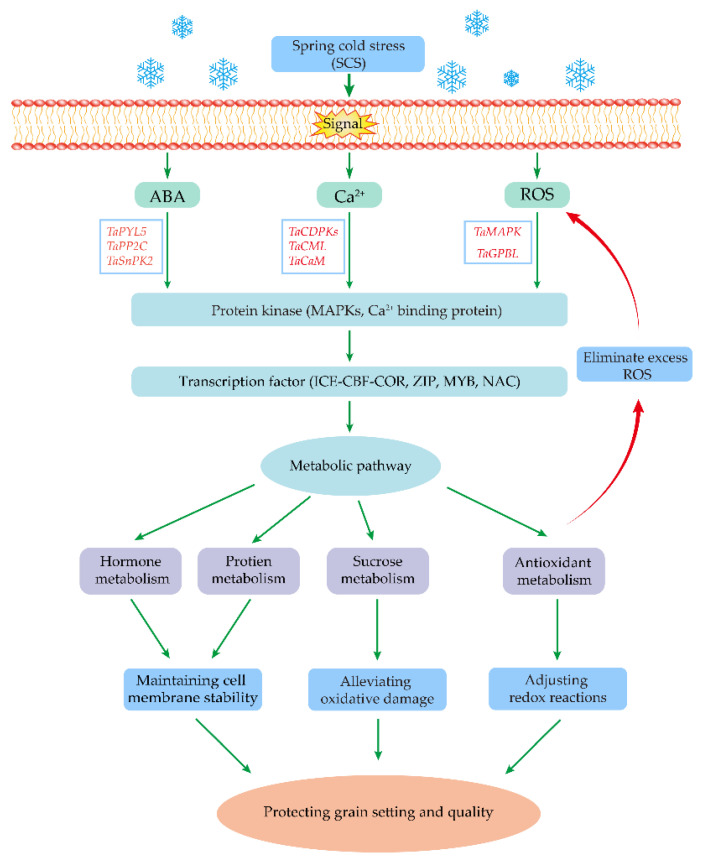
Overview of wheat responses to spring cold stress, which induces several protective measures to regulate grain setting and quality. Firstly, cold stress triggers multiple channel activation leading to the increased ABA, Ca^2+^ and ROS concentrations in the cytosol. The main components in the core ABA signaling transduction pathway include ABA receptor *TaPYL5*, *TaPP2C*, *TaSnRKs*, and the Ca^2+^ signaling transduction pathway include *TaCDPKs*, *TaCML*, *TaCaM*, which have a positive regulation of cold stress. Secondly, component changes in the MAPK cascade pathway were influenced by the activation of the ABA, ROS and Ca^2+^ pathway. Thirdly, cold stress response-induced signal transduction leads to the activation of multiple transcription factors, thereby regulating the metabolic hormone, protein, sucrose and antioxidant pathway. These alterations mitigate cell membrane damage and regulate intracellular osmotic balance, preventing the loss of grain yield and quality.

**Table 1 ijms-23-14099-t001:** Effects of SCS treatment at different stages on grain number per spike in wheat.

Cultivars	Period	Duration	Low Temperature	Grain Number	Drop Percentage (%)	Reference
Xinong979	Booting stage	12 h	15 °C/20 °C	42.2	-	[40]
			−3.5 °C/20 °C	24.8	41.2	
			−5.5 °C/20 °C	13.1	67.0	
Changhan58	Booting stage	12 h	15 °C/20 °C	40.7	-	
			−3.5 °C/20 °C	32.3	20.6	
			−5.5 °C/20 °C	17.5	57.0	
Yangmai15	Stem elongation stage	3 d	5 °C/10 °C	40.9	-	[41]
			−3 °C/0 °C	34.3	16.1	
Yangmai16	Jointing stage	3 d	6 °C/16 °C/11 °C	40.7	-	[35]
			−2 °C/8 °C/2 °C	40.2	1.3	
			−4 °C/6 °C/1 °C	39.2	3.6	
			−6 °C/4 °C/−1 °C	38.9	4.4	
	Booting stage	3 d	6 °C/16 °C/11 °C	40.0	-	
			−2 °C/8 °C/2 °C	37.2	7.0	
			−4 °C/6 °C/1 °C	36.4	8.9	
			−6 °C/4 °C/−1 °C	14.4	63.9	
Xumai30	Jointing stage	3 d	6 °C/16 °C/11 °C	36.4	-	
			−2 °C/8 °C/2 °C	36.2	0.6	
			−4 °C/6 °C/1 °C	36.2	0.6	
			−6 °C/4 °C/−1 °C	36.1	1.0	
	Booting stage	3 d	6 °C/16 °C/11 °C	36.4	-	
			−2 °C/8 °C/2 °C	32.5	10.7	
			−4 °C/6 °C/1 °C	31.1	14.6	
			−6 °C/4 °C/−1 °C	21.1	42.1	
XM21	Jointing stage	5 d	Approximately 8 °C lower than the ambient temperature	-	4.6–5.9	[42]
XZ24	Jointing stage	5 d	Approximately 8 °C lower than the ambient temperature	-	12.3–13.9	
DM22	Jointing stage	39 d	15 °C/20 °C	14.0	-	[36]
			5 °C/15 °C	8.4	40.0	
DM31	Jointing stage	39 d	15 °C/20 °C	21.0	0	
			5 °C/15 °C	4.0	81.0	
L8275	Jointing stage	39 d	15 °C/20 °C	19.0	0	
			5 °C/15 °C	17.0	10.5	
MO1	Jointing stage	44 d	15 °C/20 °C	10.4	0	
			5 °C/15 °C	0	100	
MO2	Jointing stage	44 d	15 °C/20 °C	13.6	0	
			5 °C/15 °C	0	100	

**Table 2 ijms-23-14099-t002:** Tolerant and sensitive wheat genotypes and their performances in response to spring cold stress.

Selected Indicators	Cold Stress Method	Growth Phase	Tolerant Genotypes	Performance of Tolerant Genotypes	Sensitive Genotypes	Performance of Sensitive Genotypes	References
Agronomic traits	Cryogenic incubator and solar thermal chamber	The anther seperation stage	Shannong 7859, Beijing 841, Jinmai 47, Xinmai 9, Yumai 49.	TSR ≥ 0.90	Neixiang 188, Zhengmai 7698, Xinong 889	TSR < 0.70	[138]
Agronomic traits	Field nature identification	1th to 15th in March	Yannong 5158, Huaimai 28, Huaimai 33, Jinan 17, Fanmai 5, Yannong 19, Xumai 35	Higher plant height (PH), larger grain number of main stem spike (GNMSS), GYPP, heavier grain yield per plant, stronger cold resistance, and better comprehensive agronomic traits	Jimai22, Huaimai22, Jinan17, Guomai9 Liangxing66, Zhoumai27, SXM208	Fewer GNMSS, lighter TKW, lower GYPP, weak cold resistance, and poor comprehensive agronomic traits	[151]
Agronomic traits	Field nature identification	5th to 7th in April	Bainong 207, Xinong 511	Low frozen spikelet rate, and high rate of seed setting of frozen spikelet	Zhengmai 366, Fengdecunmai 5	High frozen spikelet rate, and low rate of seed setting of frozen spikelet	[148]
Agronomic traits	Artificial chamber	3th in April	Yannong19	The correlation between GBSS activity, the starch content and the thousand kernel weight was highly significant	Yangmai18	The correlation between GBSS activity, the starch content and the thousand kernel weight was not significant	[52]
Agronomic traits	Artificial chamber	From pistil and stamen primordia differentiation stage to anthesis stage	Jimai22, Yannong19	Low dead stem rate and few residual spikes	Zhengmai8329, Wanmai50, Zhengmai366, Xian8	High rate of dead stems and many residual spikes	[147]
Physiological traits	Intelligent biochemical incubator	Jointing and booting stage	Taishan 6426	Photosynthesis rate (Pn), Transpiration rate (Tr) and Stomatal conductance (Gs) were decreased, and Internal CO_2_ concentration (Ci) increased	Taishan 4033, Jimai22	Pn, Tr and Gs decreased, and Ci were significantly decreased, and Ci increased significantly overall, and Fv/Fm decreased significantly	[152]
Physiological traits	Cryogenic incubator and solar thermal chamber	The young microspore stage	Young	Control the unsaturated lipid levels to maintain membrane fluidity	Wyalkatchem		[13]

## Data Availability

All of the data generated or analyzed during this study are included in this published article.

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
