# Peer review of "Physiology and Molecular Breeding in Sustaining Wheat Grain Setting and Quality under Spring Cold Stress"

_ijms, 2022, doi:10.3390/ijms232214099_

Round 1

Reviewer 1 Report

Reviewing “Sustaining grain setting and quality in winter wheat under spring cold stress: from molecular to breeding”. Find below my suggestions/comments for the improvement of the current ma manuscript:

Add cold stress effects on reproductive development of wheat

Add physio-morphological and molecular responses to cold stress during grain development and setting in wheat

Provide a schematic representation depicting the molecular mechanism of cold stress endorsement at reproductive stage wheat (stress perception, signal transduction, signaling cascades, protein-protein interaction networking, up/downregulation, and stress endorsement).

Add most resent information about modern breeding technologies (CRSIPR) for cold stress endorsement in wheat at reproductive stage.

Conclusion section is written poorly. 

Reviewer 2 Report

Dear Authors,

                I had a great opportunity to assess the review manuscript entitled: “Sustaining grain setting and quality in winter wheat under spring cold stress: from molecular to breeding” which is considered for publication in IJMS Special Issue: Molecular Research for Cereal Grain Quality. This review paper is intelligent written and are focused on valid problem of molecular breeding during winter cold stress and generally should be published. However, I also find some issues which must be updated/improved before publication. The list of needed improvements is presented below:

Minor:

-I suggest to check whole manuscript because in many places the font sizes are different in different parts of manuscript For example font size of Figure 1 is other than in figure descriptions. Other parts of manuscript has also this issue.

-Table 1 need more carefully editions I suggest to make full outline from cultivar to reference between  group of cultivars. I also suggest to reorganize data to show from earliest duration data so start from h then 3d, 5d and then 39 and 44d. This will be more logical in context of this table.

Major:

-          I suggest strongly to reformulated the aim currently is not well formulated and is too long  I suggest change to the: Molecular breeding in sustaining grain setting and quality in wheat under spring cold stress. Or other to be more informative

-          According IJMS publication rules the papers (also review) need the have precise formulate aim currently the paper has not any precisely formulated aim

-          Figure 2 is to small and too low quality because of that fact. Moreover, all vital elements on photos must be marked is little illogical to show photos without any markings

Best regards,

Round 2

Reviewer 1 Report

Thank you for you efforts to improve the manuscript. I had a few more suggestions:

1: In schematic representation: add some genes interaction networking which endorse cold stress in wheat under signaling pathways. For example: any hormonal signaling pathway activate any MAPK gene which thereby activate any WRKY gene to regulate (positive/negative) cold stress. I would request the authors to modify their schematic representation accordingly.

Reviewer 2 Report

Dear Authors,

All my comments was added and manuscript is corrected. I accept the publication.

Suncerely

Author Response

Thank you very much for your review of the manuscript entitled " Physiology and molecular breeding in sustaining wheat grain setting and quality under spring cold stress " (ijms-1944628).